# Association between Estimated Cardiorespiratory Fitness and Abnormal Glucose Risk: A Cohort Study

**DOI:** 10.3390/jcm12072740

**Published:** 2023-04-06

**Authors:** Robert A. Sloan, Youngdeok Kim, Jonathan Kenyon, Marco Visentini-Scarzanella, Susumu S. Sawada, Xuemei Sui, I-Min Lee, Jonathan N. Myers, Carl J. Lavie

**Affiliations:** 1Department of Social and Behavioral Medicine, Kagoshima University Graduate Medical School, Kagoshima 890-8520, Japan; 2Department of Kinesiology and Health Sciences, Virginia Commonwealth University, Richmond, VA 23284, USA; 3Faculty of Sport Sciences, Waseda University, Saitama 359-1192, Japan; 4Department of Exercise Science, Arnold School of Public Health, University of South Carolina, Columbia, SC 29208, USA; 5Department of Epidemiology, Harvard T. H. Chan School of Public Health, Boston, MA 02115, USA; 6Division of Preventive Medicine, Department of Medicine, Brigham and Women’s Hospital, Harvard Medical School, Boston, MA 02115, USA; 7Division of Cardiovascular Medicine, Veterans Affairs Palo Alto Health Care System, Stanford University, Palo Alto, CA 94304, USA; 8Department of Cardiovascular Diseases, John Ochsner Heart and Vascular Institute, Ochsner Clinical School, University of Queensland School of Medicine, New Orleans, LA 70121, USA

**Keywords:** estimated cardiorespiratory fitness, physical activity, prediabetes, diabetes, abnormal blood glucose, electronic health records, epidemiology, prevention, primary care

## Abstract

Background: Cardiorespiratory fitness (CRF) is a predictor of chronic disease that is impractical to routinely measure in primary care settings. We used a new estimated cardiorespiratory fitness (eCRF) algorithm that uses information routinely documented in electronic health care records to predict abnormal blood glucose incidence. Methods: Participants were adults (17.8% female) 20–81 years old at baseline from the Aerobics Center Longitudinal Study between 1979 and 2006. eCRF was based on sex, age, body mass index, resting heart rate, resting blood pressure, and smoking status. CRF was measured by maximal treadmill testing. Cox proportional hazards regression models were established using eCRF and CRF as independent variables predicting the abnormal blood glucose incidence while adjusting for covariates (age, sex, exam year, waist girth, heavy drinking, smoking, and family history of diabetes mellitus and lipids). Results: Of 8602 participants at risk at baseline, 3580 (41.6%) developed abnormal blood glucose during an average of 4.9 years follow-up. The average eCRF of 12.03 ± 1.75 METs was equivalent to the CRF of 12.15 ± 2.40 METs within the 10% equivalence limit. In fully adjusted models, the estimated risks were the same (HRs = 0.96), eCRF (95% CIs = 0.93−0.99), and CRF (95% CI of 0.94−0.98). Each 1-MET increase was associated with a 4% reduced risk. Conclusions: Higher eCRF is associated with a lower risk of abnormal glucose. eCRF can be a vital sign used for research and prevention.

## 1. Introduction

Cardiorespiratory fitness (CRF) is a complex trait influenced by heritability, environmental, and behavioral factors [1]. It demonstrates one’s collective physiological ability to perform aerobic activities, exercise, or sports at varying intensities and duration. It is an indicator of overall health that predicts some health outcomes better than traditional risk factors [1]. Meta-analyses demonstrated that for every metabolic equivalent (1MET = 3.5 mL·O_2_*kg^−1^·min^−1^) increase in maximal CRF, there is an 11% reduction in all-cause mortality and incidence of heart disease [2]. Given the evidence, scientists and practitioners have called for CRF to be applied as a clinical vital sign and included in public health guidelines [1,2,3,4].

Objectively measured CRF is typically determined by maximal exercise testing using standardized protocols administered by clinical staff for diagnostic and prognostic purposes [5]. Due to clinical guidelines, economics, and operational costs, regular CRF testing for non-diagnostic purposes in general practice or primary care settings is infeasible [4]. Specifically, the US Preventive Services Task Force does not recommend routine exercise testing for asymptomatic patients [6]. As a result, algorithms have been developed to estimate CRF (eCRF). eCRF typically includes parameters such as self-reported physical activity (PA) status, body mass index (BMI), and age [7]. Despite recall bias’s limitations, PA status is one of the most heavily weighted parameters in eCRF [7]. According to a recent meta-analysis, an increase in eCRF of 1 MET is associated with a 17% reduction in the general population’s risk of cardiovascular and all-cause mortality [8]. It is a better predictor than PA. Additionally, CRF was a marginally better predictor than eCRF [7,8]. Using eCRF from electronic health records (EHRs) to conduct retrospective population health investigations or health informatics is limited because PA measurement is more complex and is not universally standardized or documented [7,9,10].

EHRs provide a wealth of real-world data that can impact health at the individual and population levels. Weiskopf et al. state, “While the prospective collection of data is notoriously expensive and time-consuming, the use of EHRs may allow a medical institution to develop a clinical data repository containing extensive records for large numbers of patients, thereby enabling more efficient retrospective research” [11]. To overcome the PA data limitation in calculating eCRF from EHRs, Sloan et al. recently developed a nuanced eCRF without using PA as an algorithm parameter [12]. The eCRF algorithm included vital signs commonly found in EHRs (e.g., resting heart rate, systolic blood pressure, diastolic blood pressure) and was compared with measured CRF in 42,676 adults (21.4% females). The balanced accuracy for detecting unfit individuals ranged from 75% to 82%. According to a recent eCRF review conducted by Wang et al., there is limited research that has been conducted on eCRF with PA and health outcomes, primarily on mortality [7]. The authors also reviewed eCRF without PA and identified only a few studies that provided correlations and standard error estimates. The simple equations used only BMI and age and were found to have poor validity, and none have been validated against health outcomes [7,13].

Prediabetes and diabetes mellitus (DM) are health outcomes that cause significant death and disability worldwide [14]. The recent National Diabetes Statistical Report from the Centers for Disease Control found that 38.0% of all American adults have prediabetes, and DM prevalence is estimated at 11.3% [15]. Prediabetes rates were similar across racial and ethnic groups, all educational levels, and higher among men. Two recent meta-analyses found an independent association between CRF and DM incidence [16,17]. Investigators further indicated that relatively small increases in CRF are associated with clinically meaningful reductions (8%) in DM incidence. The authors estimated that 4% to 21% of new annual DM cases could be prevented if CRF improved by 1 MET per person [17]. While there have been no eCRF studies regarding prediabetes risk, two studies have shown that low CRF (lowest quintile) is associated with higher prediabetes incidence independent of body composition [18,19].

No studies have investigated eCRF without PA with a health outcome. Based on the evidence, we hypothesize that higher CRF will predict a reduction in abnormal glucose risk in apparently healthy adults from baseline. However, it is unclear how eCRF will compare to CRF. Therefore, our investigation aimed to determine if eCRF is a valid predictor of abnormal blood glucose and to what degree compared to CRF.

## 2. Materials and Methods

### 2.1. Study Population

Aerobics Center Longitudinal Study (ACLS) was established in 1970 and is a prospective cohort study designed to investigate the association of CRF with mortality and morbidity in adults. The total N = 43,356 ACLS cohort is based on men and women undergoing preventive medical exams at the Cooper Clinic (Dallas, TX). Patients were self-referred or referred by their physician or company. All patients provided written informed consent to participate in the study, which is reviewed and approved annually by the Cooper Institute Institutional Review Board. Study participants were primarily Caucasian with tertiary education employed in executive or professional positions. In 1979 all patients began to receive waist girth measurements. A total of 17,954 participants were included in this study (Appendix A), identified as having measured waist girth and all the eCRF parameters (age, BMI, resting heart rate, blood pressure, smoking status) at baseline. To establish an apparently healthy cohort at baseline, we excluded participants per the American Diabetic Association guidelines with known or diagnosed (fasting) diabetes (*n* = 5755), prediabetes (*n* = 531), CVD (*n* = 90), cancer (*n* = 282), abnormal ECG (*n* = 503), BMI < 18.5 (*n* = 499), age <20 or >90 (*n* = 428), chronotropic incompetence (*n* = 1253), missing information (*n* = 11) [12,20,21,22]. These inclusion/exclusion criteria (Appendix A) resulted in 8602 apparently healthy individuals (17.8% women) aged 20 to 81 years old at baseline, followed between 1979 and 2006. 

### 2.2. Clinical Examination

The Cooper Clinic preventive health exam procedure is detailed in previous publications [12,21]. After a minimum 12 h overnight fast, participants had thorough medical examinations, including resting electrocardiography, anthropometric measures, blood pressure readings, and a blood test. On a treadmill, participants underwent a test of their maximum exercise tolerance. The evaluation included the completion of a self-administered personal and family medical history. Between 1980 and 2006, follow-up exams were conducted periodically after baseline exams between 1979 and 2005.

### 2.3. Cardiorespiratory Fitness

CRF was quantified as the duration of a symptom-limited maximal graded exercise treadmill test using a modified Balke protocol in a clinical setting [21,23]. Participants were encouraged to give maximal effort; the test endpoint was volitional exhaustion or termination by the physician for medical reasons. Absolute maximal METs from the final treadmill speed and grade were calculated using metabolic equations [24]. The Balke-graded exercise test is highly correlated (*r* = 0.94) with maximal graded cardiopulmonary exercise testing [24]. 

### 2.4. Estimated Cardiorespiratory Fitness

We previously validated eCRF without PA in the ACLS baseline cohort (N = 42,676) [12]. Initially, the parameters were chosen based on the types of vital signs and clinical measures universally entered into EHR systems. After further analyses, it was determined that age, height, weight, BMI, resting heart rate, systolic and diastolic blood pressure, and smoking status were all factors that optimized the eCRF prediction algorithms. Specifically, we conducted independent linear regression analyses for men and women to generate eCRF based on the nonlinear augmentation of the predictor variables using supervised machine learning. The long-form algorithms are provided in (Appendix A, Men’s algorithm and Women’s algorithm), and we also offer a Google Sheet for researchers to calculate eCRF (ACLS eCRF Algorithm). For male participants, the multiple correlation coefficient (R) was 0.70 (mean deviation 1.33 METS), and for female participants, it was 0.65 (mean deviation 1.23 METS). For this study, after applying the eCRF algorithms, baseline max METs for eCRF and CRF were analyzed as continuous variables in the analyses. 

### 2.5. Ascertainment of Abnormal Glucose

All patients followed fasting requirements at baseline and follow-up according to the American Diabetes Association, and serum samples were analyzed for glucose using standardized, automated bioassays per the CDC Lipid Standardization Program [25]. The incidence of abnormal glucose was determined during follow-up examinations. The follow-up time for each patient was counted from the baseline examination to the first follow-up event of abnormal glucose or the last follow-up observation through 2006 in adults who did not develop either condition. The American Diabetic Association defines prediabetes (impaired fasting glucose) and DM as fasting plasma glucose concentrations of 100 to 125 and ≥126 mg/dL, respectively [25]. Those who self-reported DM or hypoglycemic medication during a follow-up were also classified as having abnormal glucose. 

### 2.6. Statistical Analysis

Descriptive statistics were calculated for study variables using mean and standard deviation (SD) for continuous variables and frequency and percentage (%) for categorical variables. Continuous variables with non-normal distribution were log-transformed before the analyses. The between-group differences in eCRF and CRF by the follow-up diabetes status (normal, prediabetes, and DM) were tested using a general linear model with a Tukey’s post hoc comparison. 

Agreement of the eCRF with CRF was evaluated using the two one-sided tests (TOST) based equivalence test, Pearson correlation (*r*), and mean absolute percent error (MAPE). Equivalency between CRF and eCRF was claimed if the 90% confidence interval (CI) for the geometric mean ratio fell within the 10% equivalence limit (0.9 and 1.11) [26]. Pearson *r* coefficient was interpreted as a weak (<0.5), moderate (0.5−0.7), or strong (>0.7) association [27]. MAPE was considered excellent (<10%), good (10%–<20%), reasonable (20%–<50%), and unacceptable (≥50%) [28]. 

Cox proportional hazard regression models were constructed predicting the risk of developing abnormal glucose (prediabetes and diabetes) based on either eCRF or CRF as a primary exposure variable before and after adjusting for covariates (age, sex, exam year, waist girth, heavy drinking, smoking, and family history of DM, high-density lipoprotein cholesterol (HDL-C), glucose, and triglycerides). The model performance was compared using the model fits statistics, including Akaike Information Criterion (AIC), Schwarz Bayesian information criterion (SBIC), Schemper and Henderson (S-H) predictive inaccuracy, and Harrell’s concordance index (C-index). AIC and SBIC are relative measures of goodness of fit for a model, where a smaller value of AIC and SBIC indicates a better model-data fit. S-H predictive inaccuracy and Harrell’s C-index are measures of the model’s predictive accuracy, with a smaller S-H predictive inaccuracy and a higher Harrell’s C-index indicating better accuracy in predicting the risk of the outcome. The proportional hazard assumption was checked by testing an interaction term of each covariate with log-transformed survival/censored time. Statistical significance was set at *p* ≤ 0.05, and SAS v9.4 (SAS Institute, Cary, NC, USA) was used for statistical analyses.

## 3. Results

Descriptive statistics of the sample characteristics are presented in Table 1. Of 8602 participants at baseline (mean age = 43.04 ± 8.94 years; male = 82.19%), 37.78% and 1.92% developed prediabetes (*n* = 3250) and DM (*n* = 165), respectively, during an average of 4.87 years of follow-up. There were significant between-group differences in CVD risk factors and family history of DM by the follow-up abnormal glucose status (Ps < 0.05). At follow-up, individuals with normal glucose levels generally showed favorable attributes compared to those with abnormal glucose.

The average CRF was 12.15 ± 2.40 METs, equivalent to the eCRF of 12.03 ± 1.75 METs within the 10% equivalence limit (Table 2). There were strong correlations (r ranges 0.69–0.75) and good agreement between the CRF and eCRF based on MAPE, ranging between 10.99% (95% CI = 10.67, 11.30) and 11.28% (95% CI = 11.03, 11.53) across the follow-up abnormal glucose status. 

Table 3 presents the results from the Cox proportional hazard regression models predicting the risk of abnormal glucose per eCRF and CRF MET increase before and after adjusting for study covariates. All models significantly associated higher eCRF and CRF with a lower risk of developing abnormal glucose levels (Ps < 0.05). The estimated hazard ratios (HRs) for eCRF and CRF were 0.96 (95% CI ranged from 0.94−0.97 for eCRF and 0.95−0.97 for CRF) without adjustment of study covariates (model 0). In the fully adjusted model (model 3), the estimated risks were the same (HRs = 0.96), but the 95% CI was slightly wider for the eCRF (95% CIs = 0.93−0.99) when compared to the measured CRF with 95% CI of 0.94−0.98. The model performance was not comparable, with fit statistics being better for the models with CRF, except for model 1. This is evidenced by lower AIC and SBIC, well as higher Harrell’s C-index, when compared to the models with eCRF.

## 4. Discussion

CRF predicts abnormal glucose but measuring CRF in primary care settings and recording PA in EHRs is not standard practice. To overcome these limitations, we sought to determine whether eCRF calculated with vital signs and standard clinical measures could predict abnormal glucose in apparently healthy adults from baseline without using PA as an algorithm parameter. According to the models, we found that measured CRF presented better overall predictive ability and eCRF to be a practical alternative. CRF and eCRF showed independent predictive ability for abnormal glucose; each 1-MET increment was associated with a 4% lower risk of incident abnormal glucose in the overall sample. Our results align with a previous investigation that found higher CRF was associated with a lower risk of abnormal glucose (per 1 MET: HR 0.99898 [95% CI 0.99861, 0.99940], *p* < 0.01) in young adults over time and with meta-analyses that show that for every 1-MET higher CRF, there is an 8–10% decrease in DM risk [16,17,29]. Our investigation is the first to show that eCRF calculated from parameters commonly found in EHRs may be useful for predicting a health outcome.

New evidence is emerging for the association of PA-based eCRF and health outcomes other than mortality [7]. Three recent eCRF cohort studies have included DM incidence as a health outcome [30,31,32]. Notably, all three studies used the ACLS Jackson eCRF (e.g., (METs) = 21.2870 + (age × 0.1654) − (age^2^ × 0.0023) − (BMI × 0.2318) − (waist girth × 0.0337) − (resting heart rate × 0.0390) + (PA × 0.6351) − (smoking × 0.4263) initially validated with heart disease and mortality outcomes in a Caucasian population [21]. These three cohort studies shed light on whether or not eCRF validated with a specific population and health outcome applies to diverse populations and health outcomes. 

Lee et al. investigated the associations of eCRF in older adults (61.5 y) with subclinical atherosclerosis, arterial stiffness, incident cardiometabolic disease, and mortality [30]. Using data from the Framingham Offspring cohort (*n* = 2962), the highest tertile eCRF was associated with a 62% lower risk of incident DM after a 15-year average follow-up. One of the study’s limitations was that PA data were unavailable for all the examination years. Other eCRF DM studies were conducted by Zhao et al. and Cabanas-Sanchez et al. in Asian cohorts. Zhao et al. found that the fittest eCRF quartile had a 58% reduction in the risk of incident DMs in rural Chinese (*n* = 11,825; 52% women) middle-aged (51y) adults at a six-year average follow-up. Additionally, they found that for each 1-MET increment in CRF, the risk of DM decreased by 15% [32]. Cabanas-Sanchez et al. found evidence for eCRF to predict the incidence of DM in adults (38.5 y) from the Taiwan MJ cohort (*n* = 200,039; 50% women) [31]. Results showed that per 1 MET change increase over an average of 6.2 years, men and women decreased their risk of DM by 25% and 36%, respectively. 

The differences in our findings are likely due to the different methods, algorithms, populations, covariates, and DM outcomes. A limitation of the three studies was that domestic PA parameters were used to calculate the Jackson eCRF rather than the Jackson PA index, which may have caused misclassification. Additionally, the Jackson eCRF was not cross-validated with CRF among participants in the respective cohorts. The Framingham cohort participant age (61.5 y) is older than the ACLS cohort (43.5 y) but similar in ethnicity, whereas the Asian cohorts are more similar in participant age to ACLS but differ in ethnicity and have more female participants. The Jackson eCRF algorithm includes BMI and waist girth as parameters, so including them in the statistical models as covariates may be over-adjusting [21]. Despite the limitations of the three studies, the Jackson eCRF was adaptable to diverse populations for predicting DM risk. Our findings show that the ACLS eCRF without PA predicted abnormal glucose risk. It is currently unknown if it applies to diverse populations as the ACLS Jackson eCRF did. 

Like other non-diagnostic tests (e.g., blood pressure, cholesterol, BMI), eCRF is potentially suitable for risk prediction and primary prevention [3,7]. Gray et al. demonstrated that low eCRF (calculated by resting heart rate, PA, and age) was useful for identifying high DM risk in females compared to five clinically validated DM risk assessment tools used in different countries [33]. Moreover, Ross and Myers have asserted that using eCRF for net reclassification improvement (NRI) may be essential [4]. NRI is a statistical technique to assess how much a given biomarker (e.g., eCRF) contributes to existing risk assessment tools to predict health outcomes. For example, the discrimination improved by ~4% when NRI was used for reclassifying Framingham risk prediction for CVD using the ACLS Jackson equation in a southern multiethnic population in Southern Xinjiang, China [34]. Similarly, Stamatakis et al. found a 7% improvement in CVD mortality in a large multiethnic population [35]. Whether or not eCRF without PA can be used to augment DM prediction tools is unknown. 

An advantage of our eCRF model is that it does not require PA assessment, providing an algorithm to rapidly auto-populate data fields and standardization across EHR systems, decreasing administration time in busy clinics. At the same time, it may allow practitioners to counsel patients on healthy lifestyle changes. While eCRF has potential as a public health tool, eCRF is not a diagnostic test. The function of clinical exercise testing goes beyond epidemiology or prevention. During exercise testing, valuable information such as the anaerobic threshold, rate of perceived exertion, heart rate response, blood pressure response, and EKG responses are documented and used for diagnosing and prognosis in CVD and respiratory diseases and developing tailored exercise prescriptions for diseased patients [5]. For example, eCRF may not be useful for patients on heart rate-lowering medications (e.g., beta-blockers). 

Although the biochemical mechanisms of abnormal glucose are not fully understood, having a higher CRF supports better insulin dynamics and glycemic control [36]. Adult exercise training studies show that increasing CRF results in favorable changes in insulin action and peripheral skeletal muscle glucose metabolism [36]. Notably, skeletal muscle insulin action may decline years before prediabetes manifests [37]. The development of insulin resistance may also be influenced by inadequate oxidative capacity, and data suggests that higher CRF is associated with improved insulin dynamics due to enhanced mitochondrial flexibility [38]. Lastly, our eCRF algorithm may be further supported by a recent variety of genetic associations identified between CRF and resting heart rate, systolic blood pressure, diastolic blood pressure, obesity, hyperglycemia, and risk of diabetes [39]. 

Our study had limitations. The observational nature of this study may limit causal inference. Measured CRF was conducted using a Balke maximal graded exercise test at baseline. This graded exercise test equation estimates absolute max METs, but highly correlates with the gold standard VO2 max testing and has been used to predict numerous health outcomes [5]. Those identified with abnormal glucose were mostly prediabetic, but recent studies show that eCRF and CRF predict DM [16,31]. Our analysis is on a primarily Caucasian cohort, but there is emerging evidence that PA-based eCRF algorithms designed from Caucasian cohorts generalize to diverse populations [31,32]. In addition, the homogeneity of the ACLS cohort strengthens internal validity. The final limitation is that the data set included only a limited proportion of women (20%).

## 5. Conclusions

A dose-dependent relationship exists between higher eCRF and a lower risk of abnormal glucose. Using eCRF in EHRs may broadly support research, informatics, risk prediction, and prevention of DM. Further research should investigate other health outcomes (e.g., cancer and CVD), diverse populations, and NRI capabilities. 

## Figures and Tables

**Table 1 jcm-12-02740-t001:** Descriptive characteristics of the study sample by follow-up glucose status ^a^.

Characteristics	Total	Follow-Up Glucose Status			*p*-Value ^b^
		Normal	Prediabetes	Diabetes	
*n* (%)	8602 (100%)	5187 (60.30%)	3250 (37.78%)	165 (1.92%)	
Exam years ^c^	1986 (14)	1988 (16) ^†^	1983 (10)	1992 (14)	<0.001
Average follow-up years	4.87 (4.58)	5.11 (4.83) ^†‡^	4.38 (4.03) ^‡^	6.59 (5.69)	<0.001
Age (years)	43.04 (8.94)	42.45 (8.96) ^†^	43.96 (8.87)	43.14 (8.59)	<0.001
Sex (*n*, %)					<0.001
Male	7070 (82.19%)	3987 (76.87%) ^†^	2957 (90.98%) ^‡^	126 (76.36%)	
Female	1532 (17.81%)	1200 (23.13%)	293 (9.02%)	39 (23.64%)	
CVD risk factors					
Glucose (mg/dL) ^c^	93 (8)	92 (8) ^†^	94 (6) ^‡^	93 (8)	<0.001
Triglycerides (mg/dL) ^c^	94 (69)	91 (66) ^†^	99 (72)	92 (76)	<0.001
HDL-C (mg/dL)	49.6 (13.90)	50.63 (14.38) ^†^	47.93 (12.88)	50.13 (14.69)	<0.001
Systolic BP (mm/Hg)	117.16 (12.71)	116.53 (12.70) ^†^	118.14 (12.64)	117.3 (13.47)	<0.001
Diastolic BP (mm/Hg)	78.94 (9.19)	78.68 (9.21) ^†^	79.37 (9.09)	78.7 (10.01)	0.004
Waist circumference (cm)	87.65 (11.8)	86.24 (12.23) ^†‡^	89.91 (10.66)	87.25 (12.55)	<0.001
BMI (kg/m^2^)	25.09 (3.36)	24.91 (3.41) ^†^	25.37 (3.23)	25.31 (3.76)	<0.001
Smoking—Yes (*n*, %)	1116 (12.97%)	641 (12.36%)	453 (13.94%)	22 (13.33%)	0.109
Family history of diabetes—Yes (*n*, %)	512 (5.95%)	330 (6.36%) ^‡^	163 (5.02%) ^‡^	19 (11.52%)	<0.001
Heavy drinking—Yes (*n*, %)	597 (6.94%)	370 (7.13%)	217 (6.68%)	10 (6.06%)	0.655

CVD = cardiovascular disease; HDL-C = high-density lipoprotein cholesterol; BMI = body mass index; BP = blood pressure. Values are presented using mean (standard deviation) for a continuous variable and *n* (%) for a categorical variable unless otherwise specified. ^a^ Prediabetes and diabetes status is determined based on the follow-up glucose levels (i.e., 100 thru <126 for prediabetes and ≥126 for diabetes and self-reported diabetes (i.e., physician-diagnosed diabetes or insulin use). ^b^ *p*-value represents the between-group difference estimated from a general linear model or *x*^2^ test of independence for a continuous or categorical variable, respectively. ^c^ Values are median (interquartile range). Log transformation was used to test the between-group differences. ^†^ Significantly different with prediabetes in post hoc comparison. ^‡^ Significantly different with diabetes in post hoc comparison.

**Table 2 jcm-12-02740-t002:** Descriptive statistics of the measured and estimated cardiorespiratory fitness (CRFs) at baseline by follow-up glucose status.

	Total	Follow-Up Glucose Status			p-Value ^a^
		Normal	Prediabetes	Diabetes	
Mean (95% CI)					
Measured CRF (METs)	12.15 (2.40)	12.10 (2.45) ^†^	12.23 (2.31)	12.18 (2.60)	<0.001
Estimated CRF (METs)	12.03 (1.75)	11.96 (1.80) ^†^	12.17 (1.64) ^‡^	11.83 (1.95)	<0.001
Equivalence testing (TOST) ^b^					
Geometric mean ratio (90% CI)	1.004 (0.998, 1.004) *	1.003 (1.00, 1.006) *	0.997 (0.993, 1.001) *	1.02 (1.002, 1.039) *	
Pearson correlation (95% CI)	0.70 (0.69, 0.72)	0.71 (0.70, 0.73)	0.69 (0.67, 0.70)	0.75 (0.67, 0.81)	
Mean absolute percent (%) error (95% CI)	11.16 (10.97, 11.36)	11.28 (11.03, 11.53)	10.99 (10.67, 11.30)	11.01 (9.67, 12.36)	

TOST = two one-sided test; METs = metabolic equivalent tasks; CI = confidence interval. ^a^ *p*-value represents the between-group difference estimated from a general linear model. ^b^ The paired equivalence testing between measured and estimated CRFs based on TOST. ^†^ Significantly different with prediabetes in post hoc comparison. ^‡^ Significantly different with diabetes in post hoc comparison. * Significantly equivalent at 10% equivalence limits (0.90 < geometric mean ratio < 1.11) based on TOST.

**Table 3 jcm-12-02740-t003:** Comparison of the Cox proportional hazard regression models predicting the incidence of prediabetes and diabetes by CRF measures ^a^.

	Hazard Ratio (95% CI)	*p*-Value	Model Fit Statistics ^b^			
			AIC	SBIC	S-H Predictive Inaccuracy Index	Harrell’s C-Index
Model 0 ^c^						
Measured CRF (METs)	0.96 (0.95, 0.97)	<0.001	55,607.3	55,613.5	0.364	0.538
Estimated CRF (METs)	0.96 (0.94, 0.97)	<0.001	55,619.5	55,625.6	0.364	0.525
Model 1 ^d^						
Measured CRF (METs)	0.93 (0.92, 0.95)	<0.001	55,114.6	55,139.1	0.333	0.618
Estimated CRF (METs)	0.88 (0.86, 0.91)	<0.001	55,102.3	55,126.8	0.333	0.617
Model 2 ^e^						
Measured CRF (METs)	0.96 (0.94, 0.98)	<0.001	55,049.3	55,098.4	0.333	0.624
Estimated CRF (METs)	0.94 (0.91, 0.97)	<0.001	55,056.1	55,105.2	0.333	0.623
Model 3 ^f^						
Measured CRF (METs)	0.96 (0.94, 0.98)	<0.001	53,273.6	53,340.8	0.332	0.663
Estimated CRF (METs)	0.96 (0.93, 0.99)	0.016	53,287.4	53,354.6	0.332	0.662

AIC = Akaike information criterion; C-index = concordance index; SBIC = Schwarz Bayesian information criteria; S-H = Schemper and Henderson’s predictive measure. ^a^ Models predicted the risk of incidence of the combined outcome of prediabetes and diabetes. ^b^ Smaller AIC, SBIC, S-H predictive inaccuracy index, and higher Harrell’s C-index indicates a better model. ^c^ Model 0 did not adjust any covariates. ^d^ Model 1 includes age (years), sex, and log-transformed exam year. ^e^ Model 2 includes waist circumference (cm), heavy drinking, smoking, and a family history of diabetes. ^f^ Model 3 further includes baseline blood lipid levels (HDL-C, log-transformed glucose, and log-transformed triglycerides levels).

## Data Availability

The datasets analyzed during the current study are not publicly accessible but are available upon reasonable request from the corresponding author.

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
