# Peer review of "Association between Estimated Cardiorespiratory Fitness and Abnormal Glucose Risk: A Cohort Study"

_jcm, 2023, doi:10.3390/jcm12072740_

Round 1

Reviewer 1 Report

Review JCM Cardiovascular Disorders „Association Between Estimated Cardiorespiratory Fitness and Abnormal Glucose Risk: A Cohort Study “

Interesting paper on an interesting topic. The manuscript is well written, the results are excellently presented and discussed in the context of the literature.

However, there are two major points of criticism:

The main results of the study were first, the correlation of eCRF with CRF according to their data, and second, the correlation of eCRF with prediabetes.

As already presented by the author it is well known that physical fitness is associated with the incidence of abnormal glucose metabolism on one hand, and eCRF is highly correlated with CRF on the other hand, as shown in previous studies. Ergo the correlation of eCRF with glucose metabolism may be expected.

Moreover, the number of approximately 38% of younger patients with a mean age of approximately 43 years developing a prediabetes during 4.3 years of follow-up is higher than expected.

To my best knowledge there are three categories of prediabetes: Impaired glucose tolerance (tested by oral glucose tolerance test), increased HbA1c, and impaired fasting glucose (IFG).  I guess that the high number of prediabetes is related to high numbers of glucose, found in the blood samples, which defines IFG. Unfortunately, this definition is imprecise since patients are often in a non-fasting state at the time of blood collection. Moreover, blood tubes are important (fluorid is necessary, if using serum samples).

Author Response

We want to thank the reviewers for their valuable insights. We have gone through and addressed each point.

As already presented by the author, it is well known that physical fitness is associated with the incidence of abnormal glucose metabolism on the one hand, and eCRF is highly correlated with CRF on the other hand, as shown in previous studies. Ergo, the correlation of eCRF with glucose metabolism may be expected.

It is clarified in lines 74~81, lines 84~88, and lines 96~99.

To my best knowledge there are three categories of prediabetes: Impaired glucose tolerance (tested by oral glucose tolerance test), increased HbA1c, and impaired fasting glucose (IFG). I guess that the high number of prediabetes is related to high numbers of glucose, found in the blood samples, which defines IFG. Unfortunately, this definition is imprecise since patients are often in a non-fasting state at the time of blood collection. Moreover, blood tubes are important (fluorid is necessary, if using serum samples).

According to the CDC and NIDDK, the prevalence rate of prediabetes in the USA is also 38%.

It is clarified in lines 156~158.

Reviewer 2 Report

Thank you for the opportunity to review the paper by Sloan and colleagues demonstrating an association between estimated CRF, using parameters widely available in the electronic health record, and risk of diabetes outcomes.  While I found the paper to be of general interest, this topic is not necessarily novel, and it is not clear that an eCRF approach that purposefully excludes physical activity parameters will lead to much clinical utility.  Building up this rationale will significantly strengthen the paper.

1) Many studies in the ACLS cohort and other cohorts have generated CRF-prediction equations with or without physical activity.  It is not clear how this particular eCRF equation greatly improves upon others that already exist, especially when it has inherent limitations that the previous ACLS equations encountered (white, male, upper class, with limited generalizabilty).

2) Is the use of an eCRF as a predictor of DM significantly superior than other DM-prediction equations that use many of the same variables (i.e., age, sex, BMI, BP, cholesterol, etc)? Relatedly, by removing physical activity from the equation, the main modifiable component of CRF is not present. Therefore, if a clinician finds that their patient has an elevated risk of DM due to their eCRF levels, what message should be delivered to that patient if PA levels don't factor in (i.e., should they be prescribed beta-blockers to lower their BP and HR since those are part of the eCRF equation)?  While I certainly agree that CRF is a vital sign that should be measured in the clinic, I'm not convinced that CRF should be estimated indirectly through other vital signs/risk factors, thus placing it as secondary to these other vital signs.  Overall, a better rationale and context for estimating CRF in the first place (as opposed to other common and widely-used risk factor equations for diabetes) is needed in the introduction and discussion.

3) Please explain how you identified your "at-risk" population which is mentioned in the Results section of the abstract.  Since the ACLS has over 80,000 baseline observations, it is not well described how this got down to 8,602.  For instance, it is not explained in the paper why participants explicitly needed waist girth measurements at baseline (as opposed to other covariates in the model).  Was waist circumference somehow used to identify the at risk population, especially since a 42% incidence of abnormal glucose seems quite high in this generally health sample?  Please also comment how cases of diabetes and prediabetes were identified at baseline (e.g., self-report? measured glucose? insulin medications?) in order to exclude those individuals.

4) Please provide more details on the actual eCRF equation in the methods rather than supplementary material.  Furthermore, the citation #18, which is referenced to describe the equation, appears incomplete in the reference list.

5) Were CRF tests that did not surpass 85% of MHR excluded in this study and the algorithms?

6) Table 1.  The "N (%)" as a heading for the list of characteristics should be labeled as "Characteristic" or "Variable."  Also, I believe "MP" in the caption should be "BP" for blood pressure.

7) Table 2.  Overall, the CRF MET values seem a little high compared to other ACLS publications.  This is especially concerning if this is truly an "at risk" population where 42% developed pre-diabetes.  It is also curious that CRF and eCRF are lowest in the "normal" group at baseline despite being young, leaner, and with lower blood pressure.  The authors may want to comment on this unusual finding in the discussion.

8) Table 3.  Are the cases of prediabetes and diabetes combined when calculating the HR?  If so, please note this in the text and caption.

9)  Line 235.  Please reference these three cohort studies.

Author Response

We want to thank the reviewers for their valuable insights. We have gone through and addressed each point.

Reviewer 3 Report

Dear authors, 

To begin with, I would like to congratulate you on your work for its originality and its high quality in their methods and justification. 

It is only my intention to suggest several minor changes that would, in my opinion, enhance the quality of the work presented.  These are: 

1. Abstract section: 

- Line 25. reflect the covariates used in the analysis. 

2. Introduction section: 

- Line 85. add "in apparently healthy adults"

- Line 85. Introduce hypothesis. 

3. Material and Methods section: 

- Line 96: indicate which parameters of eCFR

- Lines 109-110: Indicate the periodicity or number of years that the follow-up was conducted 

- Estimated cardiorespiratory system part: explain why you used these parameters in the eCFR formula. 

In general: 

- It is not clear if the estimation was used based on the eCFR that the individuals have at baseline o in each follow-up period. Clarify

- Table 1 should be improved. It is hard to understand and the interpretation of it is difficult. 

Author Response

  1. Abstract section:

- Line 25. reflect the covariates used in the analysis.

Updated.

  1. Introduction section:

- Line 85. add "in apparently healthy adults”

Updated.

- Line 85. Introduce hypothesis. (Lines 96~99)

Updated.

  1. Material and Methods section:

- Line 96: indicate which parameters of eCFR  

Updated, Line 113

-Indicate the periodicity or number of years that the follow-up was conducted.

Please see lines 131 and 201.

- Estimated cardiorespiratory system part: explain why you used these parameters in the eCFR formula.

Updated; please see lines 141~147.

In general:

- It is not clear if the estimation was used based on the eCFR that the individuals have at baseline o in each follow-up period.

It is clarified in lines 113 and 154~155.

- Table 1 should be improved. It is hard to understand and the interpretation of it is difficult.

This is a standard table; we are unclear about what is requested here. 

Round 2

Reviewer 2 Report

Thank you for addressing my comments.

Author Response

Thank you for bringing our attention to this oversight.

We made some minor revisions that you asked for. 

Updated lines 191~196

AIC and SBIC are relative measures of goodness of fit for a model, where a smaller value of AIC and SBIC indicates a better model-data fit. S-H predictive inaccuracy and Harrel's C-index are measures of the model’s predictive accuracy, with a smaller S-H predictive inaccuracy and a higher Harrel’s C-index indicating better accuracy in predicting the risk of the outcome.

Updated lines 240~243

The model performance was not comparable,  with fit statistics being better for the models with CRF, except for model 1.  This is evidenced by lower AIC and SBIC, as well as higher Harrell's C-index, when compared to the models with eCRF

Updated lines 248~249

. b smaller AIC, SBIC, S-H predictive inaccuracy index and higher Harrell’s C-index indicate a better model. c

Updated line 260

eCRF and CRF showed independent predictive ability for abnormal glucose;